# Enhancing Accuracy of Flame Equivalence Ratio Measurements: An Attention-Based Convolutional Neural Network Approach for Overcoming Limitations in Traditional Color Modeling

**DOI:** 10.3390/s24216853

**Published:** 2024-10-25

**Authors:** Lukai Zheng, Tiantian Yang, Wenjia Liu, Yufeng Lai, Jiansheng Yang

**Affiliations:** 1Department of Mechanical Engineering, Nanjing Institute of Technology, Nanjing 211167, China; lukaizheng7@gmail.com (L.Z.); lwj11834@163.com (W.L.); 2Department of Electrical and Electronic Engineering, Guizhou University, Guiyang 550025, China; 3Department of Mechanical Engineering, The University of Sheffield, Sheffield S10 2TN, UK; y.lai@sheffield.ac.uk

**Keywords:** equivalence ratio, color modeling, convolutional neural network, spatial attention mechanism

## Abstract

This paper addresses the inherent limitations in traditional color modeling techniques for measuring the flame equivalence ratio (*Φ*), particularly focusing on the subjectivity involved in threshold settings and the challenges posed by uneven 2D color distribution. To overcome these issues, this study introduces an attention-based convolutional neural network (*ACN*) model, a novel approach that transcends the conventional reliance on B/G color features (*T_f_*). The *ACN* model leverages adaptive feature extraction, augmented by a spatial attention mechanism, to more effectively analyze flame images. By amplifying key features, autonomously minimizing background noise, and standardizing variations in color distribution, the ACN model in this experiment achieved a prediction accuracy of 99%, with a 76% reduction in error rate compared to the original model, significantly improving the accuracy and objectivity of flame *Φ* measurement. This method marks a substantial development in the precision and reliability of flame analysis.

## 1. Introduction

Accurate measurement of the equivalence ratio (*Φ*) in hydrocarbon flame combustion is essential for optimizing fuel conversion efficiency, reducing pollutant emissions, minimizing heat loss, and ensuring flame stability [1]. Specific radicals, such as hydroxyl (OH), methylidyne (CH), and dicarbon (C_2_), emit chemiluminescence at different wavelengths (308 nm, 430 nm, and 516.5 nm, respectively) during their transition from excited to ground states. Analyzing this chemiluminescence within a flame’s reaction zone facilitates a precise assessment of combustion characteristics, including stoichiometry and heat release [2,3]. The calibration of the chemical *Φ* using these chemiluminescence ratios mitigates signal variances caused by factors such as mass flow rate, optical depth, and obstructions in the optical path, thereby enhancing the reliability of combustion analysis [4].

The spectroscopic analysis of combustion flames, noted for its high resolution and quantitative methodology, plays a crucial role in this context. Clark’s research demonstrated that C2*/CH* ratio serves as an effective indicator of flame characteristics under specific conditions, with equivalence ratios varying from 0.6 to 1.5 [4]. Additionally, Haber’s study supported the reliability of CH*/C2* chemiluminescence ratios under comparable conditions [5].

The integration of a spectrometer with a grayscale digital camera for the purpose of capturing images of free radical chemiluminescence has emerged as a crucial tool in the analysis of 2D combustion processes. Kojima et al. [6,7] employed a Cassegrain mirror system in conjunction with a spectrometer imaging setup to detect chemiluminescence in laminar pre-mixed methane/air flames, recording OH*/CH* ratios that varied from 0.9 to 1.5.

The introduction of color charge-coupled device (CCD) digital cameras has markedly enhanced the study of combustion processes by enabling the analysis of color features (*T_f_*) in 2D flame images. In accordance with Maxwell’s trichromatic principle, these cameras capture three broadband spectral bands, which facilitate real-time measurements of free radical chemiluminescence [8]. Huang and Zhang [9,10] demonstrated the accurate representation of CH* and C_2_* chemiluminescence through blue (B) and green (G) emissions within red–green–blue (RGB) color space. However, the physical conversion relationship between the intensities of the digital color channels (IB and IG) and the chemiluminescence emissions (CH* and C_2_*) remains inadequately understood. Yang and Ma employed optical correction and mathematical methodologies to establish the conversion relationship between the average intensity ratios of B and G channels and the chemiluminescence of CH* and C_2_* [11].

In images of premixed hydrocarbon flames captured by CCD cameras, a dual flame structure is evident, consisting of an inner flame that burns in a highly premixed mode and an outer flame that consumes incompletely burned intermediates [12]. The digital flame color discrimination (DFCD) technique, which was developed to minimize the impact of the external flame on image analysis, effectively reduces the impact of diffuse flames in RGB channels through H-channel threshold segmentation [13,14]. However, the effectiveness of this method in accurately measuring *Φ* is constrained by its dependence on subjective judgment for threshold adjustments.

Furthermore, the heterogeneous distribution of intermediates in alkane flames leads to nonuniform representations of R, G, and B colors in images captured by CCD cameras [12]. This phenomenon, referred to as the ‘color thickness effect’, results in darker edge colors in comparison to the interior regions, thereby reflecting the integrated spectral response from various radicals. Conventional color modeling techniques, which primarily concentrate on individual *T_f_* such as the blue-to-green (B/G) ratio, may overlook these subtleties, potentially compromising the accuracy of *Φ* measurements [15].

Deep learning has become increasingly prevalent in the field of combustion diagnostics due to its capacity to adaptively learn multi-dimensional features [16]. Han et al. [17,18] proposed the use of neural networks and semi-supervised models that leverage deep learning techniques for the processing of flame images and the detection of steady states in combustion processes. Similarly, other researchers have employed deep learning methodologies to analyze flame images, thereby developing soft models for various measurements related to combustion [19,20,21].

This study presents an attention-based convolutional neural network (*ACN*) model designed for predicting *Φ* in methane combustion flames. This deep learning methodology surpasses conventional color modeling techniques by eliminating the need for prior knowledge in feature extraction and exhibiting improved generalization capabilities for previously unencountered combustion states [22,23]. The efficacy of deep learning is attributed to its ability to extract essential features directly from data through a generalized learning process. During the supervised fine-tuning phase, regression models are developed to elucidate the nonlinear relationships between the extracted features and *Φ*, representing a significant development in the field of combustion analysis.

## 2. Materials and Methods

### 2.1. Experimental Setup

The experimental setup (Figure 1) comprised a variety of essential equipment necessary for conducting comprehensive combustion studies. This equipment included methane gas cylinders, air compressors, manometers, flow meters, a premix tube, burners, a Photron SA-4 high-speed color CMOS camera, personal computers, and a gas tube. A key component of this configuration was the computer-operated flowmeter, which played a crucial role in accurately regulating the flow rates of fuel gas and air, thereby achieving the desired *Φ* for the mixture. Following meticulous calibration, the premixed mixture was subsequently introduced into the main lamp burner through a precisely selected nozzle with a diameter of 10 mm, which was critical for establishing optimal combustion conditions for the study.

To capture images of the flames, a Photron FASTCAM SA4 high-speed color CMOS camera (Saugus, MA, USA) equipped with cooling devices was employed to prevent damage from elevated temperatures. The camera transmitted the captured signals to a computer through an IEEE-1394a interface. Each image, characterized by a resolution of 1280 × 720 pixels and a color depth of 24 bits, was collected at a rate of one frame per second. This meticulous data collection process facilitated a comprehensive and systematic analysis of the combustion phenomena.

A total of 10 sets of images depicting methane flame combustion were compiled, with *Φ* ranging from 0.73 to 1.47. Each set comprised 100 images captured using a digital color camera, which were subsequently cropped and resized to dimensions of 256 × 256 pixels. Figure 2 presents a selection of these images, effectively illustrating the variation in *Φ*.

### 2.2. ACN

#### 2.2.1. Overall Network Structure of ACN

Deep learning algorithms serve as a fundamental component in the domain of computer vision, exhibiting remarkable effectiveness in image processing and analysis. These advanced algorithms are proficient in autonomously learning and representing a diverse array of features, which allows them to competently execute tasks such as denoising, feature extraction (e.g., color, geometry, and texture), and the management of multi-dimensional information. Their robust capabilities make these algorithms particularly effective for tasks necessitating extensive image processing and regression prediction.

The *ACN* model constitutes a notable development in the field of computer vision. This model adeptly leverages the correlation between flame imagery and *Φ*. In Figure 2, the model illustrates the color transition observed in flames, which progresses from B/G and ultimately to a reddish-yellow hue at elevated levels of *Φ*. This color variation, in conjunction with the flame’s tapered morphology and the pronounced stratification in brightness at its periphery, serves as a critical visual indicator for comprehensive analysis.

In Figure 3, the *ACN* model is organized into six stages: convolution (Conv2D), activation, max pooling (Maxpool2D), spatial attention mechanism, flattening, and dense layers. Each stage plays a crucial role in modifying the dimensions of the tensors involved in the process. The model employs mean absolute error (MAE) as its loss function and is optimized using the Adam algorithm with a batch size of 32.

The overall network structure of the *ACN* model is structured into three primary layers: the input layer, the hidden layer, and the output layer. The input layer is specifically designed to adapt dynamically to the structural requirements of the network. It processes RGB images with three channels, preserving their original dimensions of 256 × 256 pixels. Within the input layer, each image is represented as a 3D array (256 × 256 × 3), corresponding to the R, G, and B channels.

The hidden layer, which comprises steps 2 to 4 of the model’s operation, incorporates two critical components: a convolution pooling layer (CP128) and a spatial attention mechanism layer (ATT). Ultimately, the output layer, consisting of steps 5 and 6, is structured as a fully connected layer.

#### 2.2.2. CP128 Layer: Feature Detection and Refinement

CP128 serves as a critical element within the hidden layer of the *ACN* model, tasked with the processing of RGB color flame images (Figure 4).

CP128 incorporates Conv2D, activation, and MaxPool2D operations, as delineated in Equations (1a)–(1c). The initial phase of this process involves a 3 × 3 convolution operation, with the kernel parameters established through model back-training. This phase is essential for the detection of various features, including color, edges, textures, and shapes within RGB images. During the convolution process, a cross-correlation function, as described in Equation (1a), is utilized. This function combines the pixel values of the convolution kernel matrix *K* with those of the input matrix *I*, resulting in the output matrix *O*.
(1a)$$O(i,j)=(K*I)(i,j)=\sum_{m}\sum_{n}I(i+m,j+n)K(m,n)\;\;\;\;$$
(1b)$$\;\;\;\;\;\;\;\;\;\;\;\;\;\;\;\;\;\;\;\;\;\;\;\;\;f(x)=\left\{\begin{array}{c} x,x\geq 0\\ 0,x<0 \end{array}\right.=max(0,x)$$
(1c) yij=xmn(m,n)∈PIJmax

Following the convolutional operation, the CP128 layer implements a rectified linear unit (ReLU) activation function, as indicated in Equation (1b). This function serves to preserve positive values while eliminating negative ones, thereby addressing the vanishing gradient problem and facilitating faster convergence. Subsequently, a 2 × 2 pooling kernel is employed during the maximum pooling phase (Equation (1c)) to selectively filter essential data, thereby enhancing analytical capabilities and reducing the risk of overfitting. The CP128 layer, which utilizes 128 convolutional kernels, generates a tensor of dimensions 127 × 127 × 128, thereby significantly improving the model’s feature extraction efficiency.

#### 2.2.3. ATT: Focusing on Flame Features

Figure 5 presents ATT, an essential element of the hidden layer that processes multiple tensors (Table 1). ATT functions in two primary phases. In the first phase, it applies a nonlinear transformation using the hyperbolic tangent function (tanh) to the input tensors (CP128 output tensors), resulting in the generation of the first attention score tensor, referred to as Score1. Simultaneously, these input tensors are subjected to additional processing to create a ‘Diff’ tensor, which captures the absolute differences between each tensor element and the mean value of its corresponding row. Subsequently, the ‘Diff’ tensor is processed through a similar nonlinear transformation to produce Score2, the second component of the attention score matrix.

The integration of Score1 and Score2, facilitated by softmax normalization, results in the formation of a weight tensor. This tensor is subsequently merged with the image mask tensor, producing the final output tensor. Importantly, each pixel in the mask tensor corresponds to the maximum value derived from the output tensor of the preceding layer, computed on a per-pixel basis within the depth dimension. This integration is essential for enhancing significant features in flame images while minimizing background noise, thereby substantially improving the performance of the neural network.
(2a)$$C=A*B,C_{ij}=\sum_{k=1}^{p}a_{ik}b_{kj}$$
(2b)$$C=A\;\cdot \;B,\;C_{ij}=a_{ij}b_{ij}$$
(2c)$$f(x)=tanh(x)=\frac{e^{x}-e^{-x}}{e^{x}+e^{-x}}$$
(2d)$$S_{x_{i}}=\frac{e^{x_{i}}}{\sum_{j=1}^{N}e^{x_{j}}}$$

ATT incorporates various mathematical operations. Equations (2a) and (2b) involve tensors *A*, *B*, and *C* for specific tensor calculations. Subsequently, Equations (2c) and (2d) apply the tanh activation function and softmax normalization function to the point values *x* of the tensor elements.

The tanh function is essential in ATT due to its function in nonlinearly scaling input data within a range of −1 to 1, which effectively normalizes the data around a mean of zero. This normalization process is crucial for enhancing the stability and learning efficiency of the model. Simultaneously, softmax function converts attention scores into probability distributions, thereby emphasizing critical information while reducing background noise. The integration of these mechanisms substantially improves the model’s ability to analyze complex data patterns with increased accuracy and efficiency.

Additionally, the implementation of adaptive training within this layer, which encompasses the parameters W0, W1, B0, B1, and V, is essential for enhancing the model’s focus and improving its overall performance. Utilizing a back-propagation algorithm, this adaptive training methodology modifies these parameters in reaction to output errors, thereby optimizing the accuracy and efficiency of the neural network.

#### 2.2.4. Output Layer: Feature Transformation and Predictive *Φ*

Figure 6 illustrates the critical functions performed within the output layer of the network. The flatten operation plays a pivotal role in converting the feature tensor output from ATT into a 1D vector. This transformation is essential for efficiently organizing the features for the dense layer, thereby simplifying the data while preserving significant spatial structures.

The dense layer, as illustrated in the diagram of the output layer, comprises a network of interconnected neurons, with each neuron connected to all neurons in the preceding layer. This structure enables thorough feature analysis and integration, thereby facilitating the generation of accurate predictions based on a comprehensive understanding of the data. The primary function of the dense layer within the output layer involves the utilization of weight parameters and a bias term to effectively process and interpret these features, which is essential for producing accurate and precise measurements of *Φ*.

### 2.3. Traditional Color Modeling

#### 2.3.1. Analysis of Flame Image Structure

The analysis of the flame image, characterized by a *Φ* value of 1.47 (Figure 7) reveals three regions: the external radiation region, the contour edge region, and the internal combustion region. The external radiation region is marked by a broad spectrum of reddish-yellow hues. In contrast, the central target, or internal combustion area, predominantly displays a B/G hue. However, B/G hue observed in the central area of the flame is not uniform; it is impacted by the uneven distribution of reaction intermediates and the effects of spatial spectral integration. Additionally, the contour edges of the flame exhibit increased brightness, a result of spatial spectral integration, which stands in contrast to the internal combustion area. This variation underscores the complex interaction of color within the flame’s structure.

#### 2.3.2. Modulation of H Threshold in DFCD

The conventional color modeling process encompasses multiple stages to extract *T_f_* from the original image (Figure 8). The procedure initiates with Gaussian filtering, which is designed to mitigate random noise, thereby enhancing the precision of feature extraction. Subsequently, the DFCD method is employed to delineate the internal flame region by eliminating external flames. The following step entails the computation of *T_f_* as the mean ratios of the *B* and *G* channels. This calculation is essential for establishing a correlation between *T_f_* and *Φ* through cubic regression analysis, thereby optimizing the *T_f_* extraction process.
(3a)$$H_{ij}=\left\{\begin{array}{c} 1,h_{low}\leq h_{ij}\leq h_{up}\\ 0,h_{ij}<h_{low};h_{ij}\leq h_{up} \end{array}\right.$$
(3b)B=H∗b ;G=H∗g 
(3c)$$T_{f}=\frac{\sum_{i=1}^{m}\sum_{j=1}^{n}B_{ij}}{\sum_{i=1}^{m}\sum_{j=1}^{n}G_{ij}}\;$$
(3d)$$\Phi =aT_{f}^{3}+bT_{f}^{2}+cT_{f}+d$$

The DFCD method, which is designed to enhance sensitivity, encounters difficulties in effectively mitigating flame background interference. This method encompasses specific color segmentation procedures, as delineated in Equations (3a)–(3d). Notably, Equation (3a) emphasizes the subjective aspect of adjusting the upper and lower thresholds of the H component within the H layer. Figure 9 demonstrates the implications of these adjustments on background radiation. Altering the upper limit to values such as 0.53, 0.60, and 0.67 yields different results. An excessively low upper limit results in the loss of the target flame image (Figure 9a), while an excessively high upper limit fails to adequately eliminate background radiation (Figure 9c). Consequently, it is imperative to tailor the upper and lower limit parameters for each set of *Φ* images, taking into account their different characteristics.

## 3. Results

### 3.1. Simulation Curve of Traditional Color Modeling

The efficacy of the DFCD method in eliminating background radiation from the outer flame at various values of *Φ* is well documented. A comparison between the original image and the image subsequent to the removal of background radiation is presented in Figure 10.

Figure 11 depicts the variations in B/G ratios obtained from flames in relation to *Φ*, both prior to and following the application of DFCD treatment. Before the treatment, the B/G values display a disordered pattern. However, subsequent to the treatment, they demonstrate a closer alignment with the cubic regression fitting curve. Despite this enhancement, significant discrepancies remain evident, particularly during the rich combustion phase.

In the analysis of flame images, traditional color modeling’s H-color segmentation threshold exhibits a high sensitivity to the removal of background radiation, necessitating customized threshold settings for each set of flame images based on the parameter *Φ*. While this approach effectively mitigates background radiation, it results in inconsistent B/G ratios across different images, which negatively impacts prediction accuracy. Despite the removal of background radiation following DFCD processing, the extraction of B/G *Tf* continues to be impacted by the irregular distribution of 2D color and edge thickness. Although the variations in the B/G ratio and *Φ* predominantly adhere to regression curves, the accuracy of their fitting is suboptimal, indicating a need for further refinement. This paper proposes the application of an *ACN* model to address these challenges, with the objective of significantly enhancing the accuracy of the model.

### 3.2. Evaluating ACN Performance: Ablation Study Insights

A series of ablation experiments were conducted to evaluate the impact of various structural configurations on the performance of the ACN model. The dataset was divided into training and testing subsets in a 7:3 ratio, with the objective of predicting the flame *Φ* from 2D images. Following ten training cycles, the *R*^2^ coefficient, MAE, and their respective variances were calculated to assess prediction accuracy and stability.
(4a)$$R^{2}=1-\frac{\sum_{i=1}^{N}{\left(y_{i}-\hat{y_{i}}\right)}^{2}}{\sum_{i=1}^{N}{\left(y_{i}-\bar{y}\right)}^{2}}\;\;$$
(4b)$$MAE=\frac{\sum_{i=1}^{N}\left|y_{i}-\hat{y_{i}}\right|}{N}\;$$
(4c)$${S_{x}}^{2}=\frac{\sum_{i=1}^{N}{\left(x_{i}-\bar{x}\right)}^{2}}{N}\;$$

Equations (4a)–(4c) present the formulas for calculating *R*^2^, *MAE*, and the variance, where $\hat{y_{i}}$ is the true value, $\bar{y}$ is the mean value of the variable, and *N* is the total number of elements of the variable.

The experiment incorporated four model structures (Figure 12a). Model 1 was constructed without CP128, Model 2 omitted ATT, and Model 3 included these layers in reverse order. The uniform input across all models consisted of a methane flame image with *Φ* of 1.47. Figure 12b depicts the transformation of this flame image by each model variant.

In Figure 12 and Table 2, two significant conclusions can be derived as follows:

The exclusion of either CP128 or ATT adversely impacts the performance of the model. Relying exclusively on ATT results in inadequate feature characterization for *Φ*, even though it partially mitigates background radiation. Conversely, utilizing only CP128 improves feature extraction but undermines the stability of the model due to the persistence of residual background radiation.

The strategic configuration of model layers is of paramount importance. Positioning ATT after the CP128 layer significantly enhances the model’s accuracy and stability, effectively reducing background radiation. Conversely, positioning ATT prior to the CP128 layer results in suboptimal results, primarily due to insufficient removal of background radiation.

## 4. Discussion (Comparative Analysis: *ACN* Model Versus Traditional Color Model)

The effectiveness and superiority of the *ACN* model in feature extraction are illustrated in Figure 13, which depicts the variations in *MAE* and *R*^2^ values across training batches. It demonstrates a consistent reduction in *MAE* and an increase in R^2^, signifying the model’s convergence towards optimal performance. The model’s ability to adaptively modify training parameters plays a crucial role in this enhancement, thereby improving its capacity to accurately represent the nonlinear relationship between flame imagery and *Φ*.

In this experiment, the ACN model was trained using the Adam optimizer for adaptive adjustment. Throughout the training process, the model was divided into 10 epochs, with each epoch employing a mini-batch learning approach that processed 32 samples at a time. This strategy ensured the rational utilization of computational resources while enhancing training efficiency and ensuring stable updates of model parameters. The Adam optimizer combines the advantages of momentum and RMSprop optimization algorithms, using a larger learning rate in the early stages of training to facilitate rapid learning; as the optimization phase approaches, the learning rate gradually decreases, ensuring adaptive adjustment of each parameter, thereby improving the efficiency and accuracy of model training.

Figure 13 presents the changes in the coefficient of determination (*R*^2^) and the mean absolute error (MAE) during the model training process as a function of the number of epochs. The improvement in model performance can be analyzed from two key points of change. For the initial rapid adaptation, in the early stages of training, the *R*^2^ value rapidly increases, and the MAE value rapidly decreases, indicating that the model quickly adapts to the data characteristics and improves prediction accuracy within the first four epochs. This phenomenon benefits from the dynamic learning rate adjustment strategy of the Adam optimizer, which enables the model to rapidly adjust its parameters and quickly capture patterns in the data during the initial stages. For the convergence and stability, as the number of epochs increases, the *R*^2^ and MAE curves gradually stabilize and ultimately converge. This stable trend indicates that the model’s performance has approached its upper limit and that there is no evidence of overfitting or curve oscillation. The smooth convergence of the curves suggests that the model’s training process is stable, especially with the help of the Adam optimizer, which effectively avoids overfitting and ensures the model’s generalization ability to unknown data.

Figure 14a identifies flame images in terms of three primary regions: external radiation, contour edge, and internal burning. Figure 14b evaluates traditional color modeling techniques, illustrating their effectiveness in mitigating the impacts of external radiation. In contrast, Figure 14c provides a 2D visualization of the features extracted by the *ACN* model. This representation highlights the *ACN* model’s ability to reconstruct images, effectively eliminating external noise while preserving critical flame characteristics and maintaining the geometric integrity of the contour edge regions. This demonstrates the model’s advanced capabilities in the analysis of flame images.

The efficacy of the *ACN* model is further substantiated by the observed parameter changes during the training process (Figure 13) and the reconstructed feature maps (Figure 14). This model effectively extracts critical features from flame images, facilitating accurate predictions of the flame *Φ* through linear transformations in the output layer. Figure 15 illustrates a comparison between the actual flame *Φ* and the predictions made by both traditional color modeling and the *ACN* model. The predictions generated by the *ACN* model demonstrate a closer alignment with the actual values during the rich combustion phase, thereby mitigating the erratic fluctuations typically associated with traditional color models.

The performance evaluation metrics summarized in Table 3 further validate the improvements of the attention-based convolutional neural network (ACN) model compared to traditional color modeling. In terms of the coefficient of determination (*R*^2^), the ACN model exhibits an increase from 0.9375 to 0.9958, demonstrating a better ability to explain data variability. Concurrently, the mean absolute error (*MAE*) of the model’s predictions decreases from 0.0504 to 0.0122, enhancing the accuracy of equivalence ratio predictions.

The low values of the variance of *R*^2^ (*S_R_*_2_^2^) and the variance of MAE (*S*_MAE_*^2^*) for the ACN model collectively demonstrate the model’s robust performance. The low values of these two variance coefficients indicate that the ACN model maintains high stability and low error fluctuation across 10 independent test results.

## 5. Conclusions

This research aims to advance the methodology for predicting flame *Φ*, thereby enhancing the efficiency and accuracy of real-time combustion diagnostics. Traditional methods, which rely on image filtering and DFCD, have faced challenges such as the necessity for subjective adjustments and the reliance on prior knowledge for feature extraction. These methods often suffer from reduced accuracy due to the complexity of 2D color distribution in flame images. To overcome these limitations, this paper introduces an innovative model mechanism.

The proposed *ACN* model constitutes a substantial improvement over conventional techniques. By utilizing deep learning for the extraction of critical features from flame images and subsequently developing a regression model to accurately measure flame *Φ*, the *ACN* model demonstrates superior performance. Its sophisticated design enables the efficient processing of extensive image data and the extraction of nonlinear features with greater effectiveness compared to traditional methods. Experimental results confirm that the *ACN* model surpasses existing approaches in both image analysis and combustion diagnostics.

A key aspect of this study is the controlled maintenance of a fixed camera distance during flame image capture. However, given that the relative positions of the camera and flame may vary, the model must adapt accordingly. Convolutional networks, known for their robust image processing capabilities, empower the *ACN* model to accommodate such variations in camera and flame positioning. Future enhancements of the model are anticipated, with the objective of deploying it in industrial settings to further improve the accuracy and efficiency of combustion diagnostics.

## Figures and Tables

**Figure 1 sensors-24-06853-f001:**
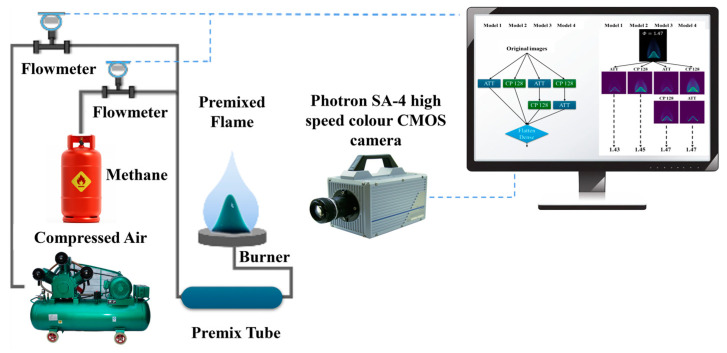
Schematic of experimental setup.

**Figure 2 sensors-24-06853-f002:**
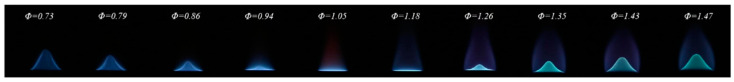
Methane flames with *Φ* ranging from 0.73 to 1.4.

**Figure 3 sensors-24-06853-f003:**
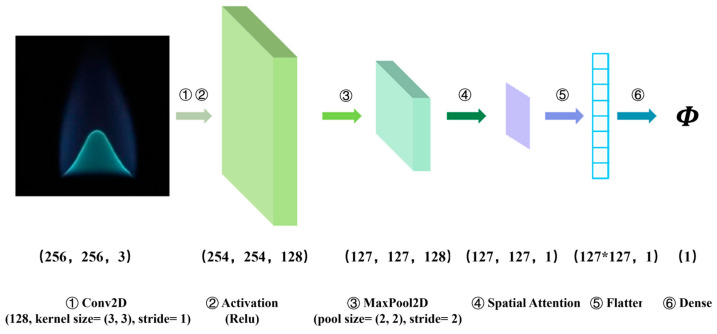
Structure of *ACN* model from input to output layers.

**Figure 4 sensors-24-06853-f004:**
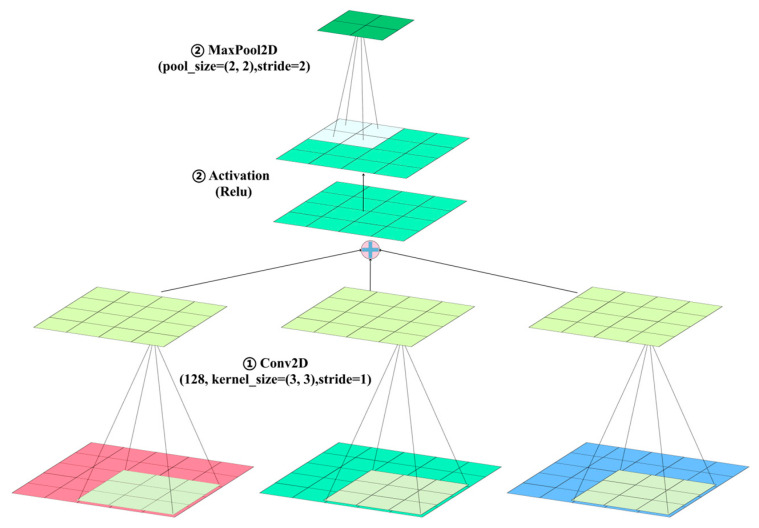
Schematic of Conv2D, activation, and Maxpool2D processes.

**Figure 5 sensors-24-06853-f005:**
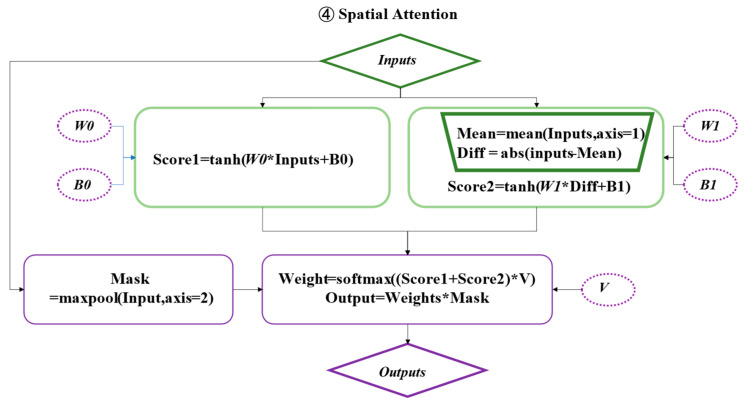
Flowchart of spatial attention mechanism.

**Figure 6 sensors-24-06853-f006:**
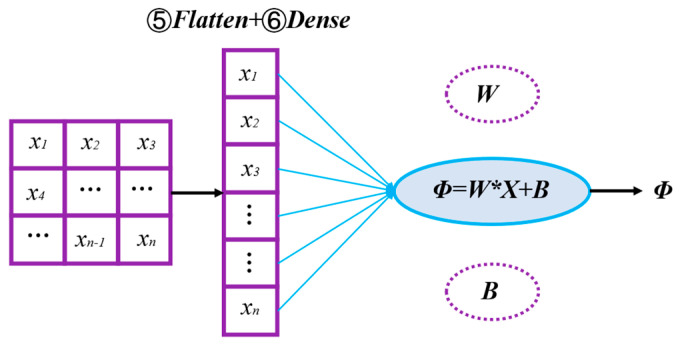
Schematic of output layer with flatten and dense operations.

**Figure 7 sensors-24-06853-f007:**
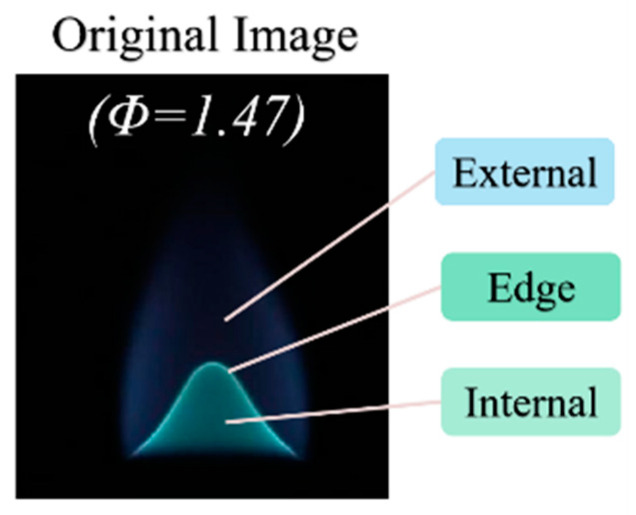
Schematic of background radiation and marginal internal differences in a flame with an *Φ* of 1.4.

**Figure 8 sensors-24-06853-f008:**
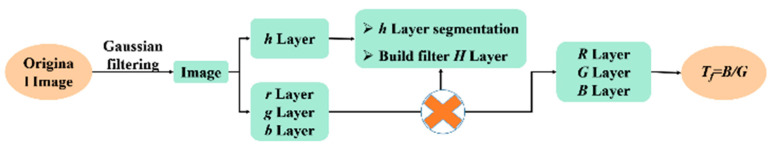
Flowchart of *T_f_* extraction using color modeling method.

**Figure 9 sensors-24-06853-f009:**
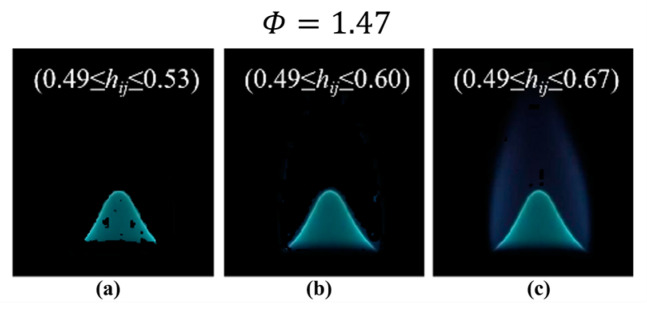
Segmented flame images with different H thresholds. (**a**) H threshold 0.49 to 0.53; (**b**) H threshold 0.49 to 0.6; (**c**) H threshold 0.49 to 0.67.

**Figure 10 sensors-24-06853-f010:**
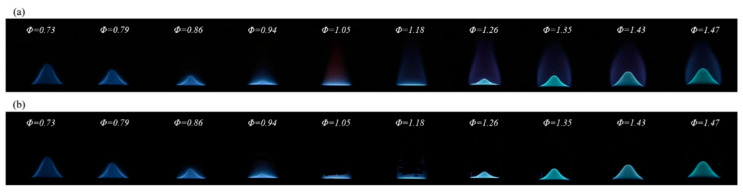
Comparison of flame images before and after DFCD processing. (**a**) Original flame images; (**b**) DFCD processed flame images.

**Figure 11 sensors-24-06853-f011:**
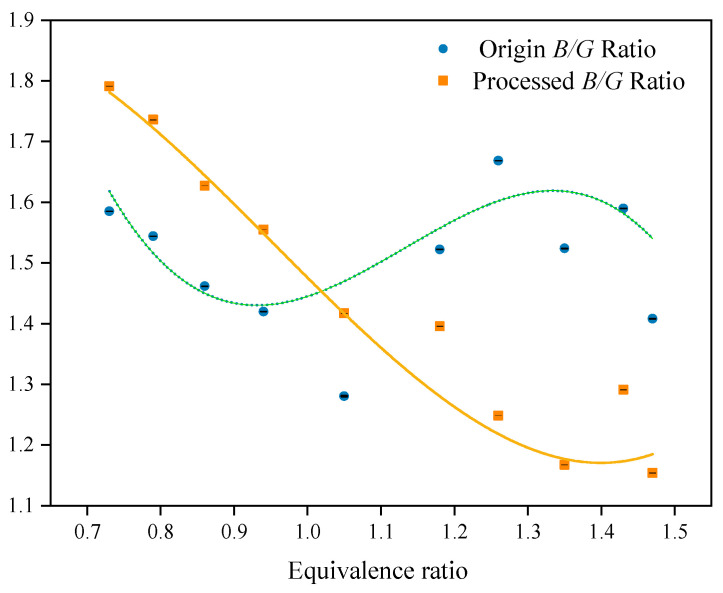
Plot of extracted *Tf* (B/G) with *Φ* before and after DFCD treatment.

**Figure 12 sensors-24-06853-f012:**
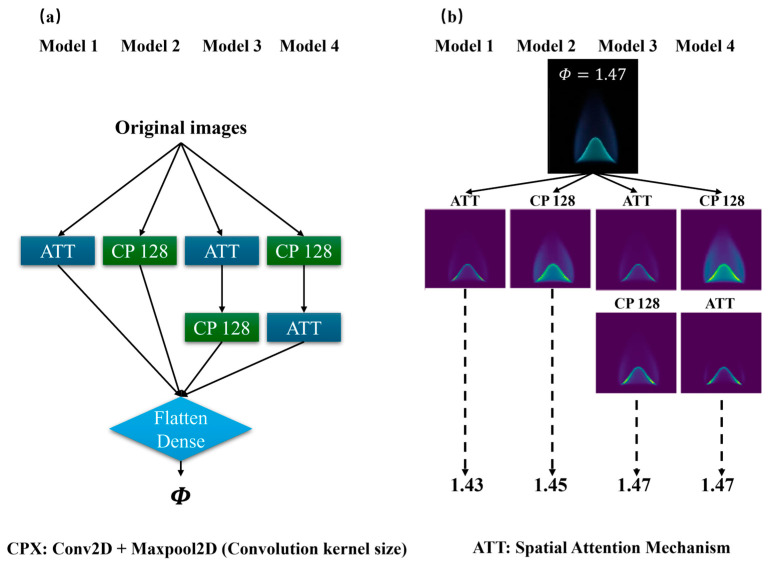
(**a**) Structure of ablation experimental models; (**b**) Network image transformation for methane flame image at *Φ* of 1.47.

**Figure 13 sensors-24-06853-f013:**
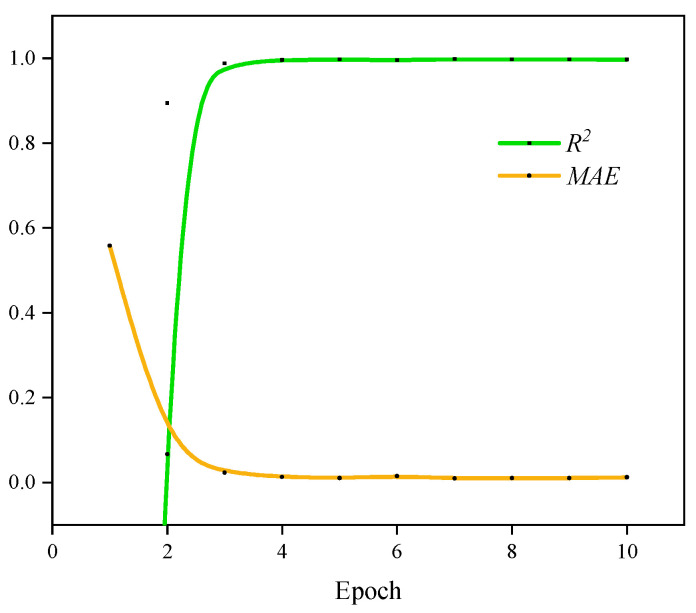
R^2^ and MAE change curves with epochs.

**Figure 14 sensors-24-06853-f014:**
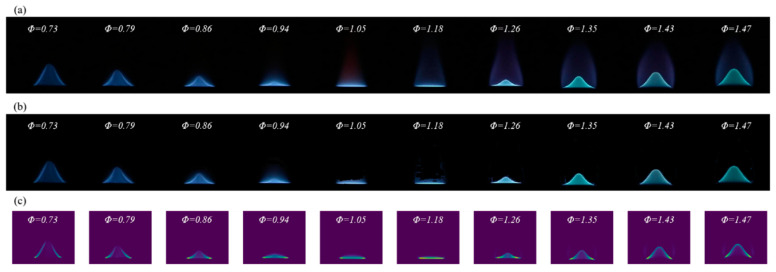
Comparison of original images processed by DFCD and ACN model. (**a**) Original flame images; (**b**) DFCD processed flame images; (**c**) ACN model training processed flame images.

**Figure 15 sensors-24-06853-f015:**
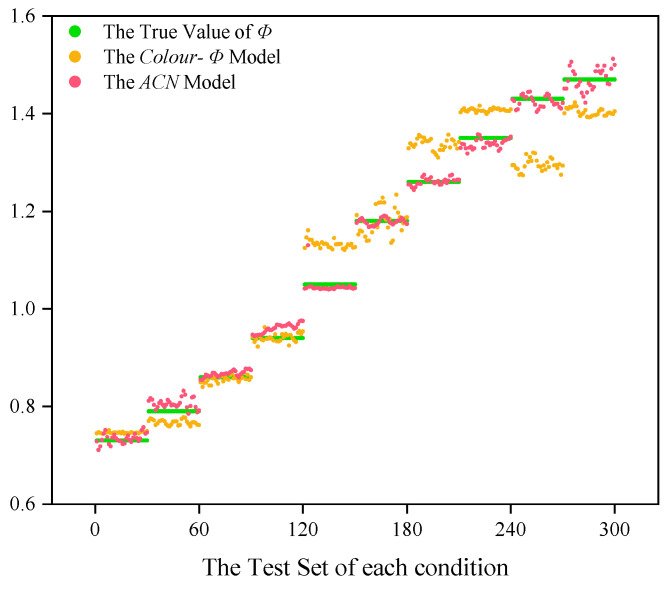
Comparison of actual *Φ* with predictions from traditional color model and *ACN* model.

**Table 1 sensors-24-06853-t001:** Tensor and dimension in spatial attention mechanism.

Tensor	Dimension
Inputs	(127, 127, 128)
Mean	(127, 1, 128)
Diff	(127, 127, 128)
W0, W1	(128, 32)
B0, B1	(127, 127, 32)
Score1, Score2	(127, 127, 32)
V	(32, 1)
Weight	(127, 127, 1)
Mask	(127, 127, 1)
Output	(127, 127, 1)

**Table 2 sensors-24-06853-t002:** Evaluation parameters for ablation experiments.

Model Structure	*R^2^*	*S_R2_^2^*	*MAE*	*S_MAE_^2^*
*ATT*	0.9176	1.20 × 10^−4^	0.0610	1.8 × 10^−5^
*CP128*	0.9763	1.06 × 10^−4^	0.0272	2.38 × 10^−5^
*ATT*-*CP128*	0.9809	7.56 × 10^−6^	0.0200	8.52 × 10^−6^
*CP128*-*ATT*	0.995	818 × 10^−8^	0.0122	3.45 × 10^−7^

**Table 3 sensors-24-06853-t003:** Evaluation parameters of traditional color model and *ACN* model.

Model	*R^2^*	*S_R2_^2^*	MAE	*S* _MAE_ * ^2^ *
*Color-Φ* model	0.9375	1.22 × 10^−5^	0.0504	1.49 × 10^−6^
*ACN* model	0.9958	8.18 × 10^−8^	0.0122	3.45 × 10^−7^

## Data Availability

All research data are in the paper.

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
