# Peer review of "Enhancing Accuracy of Flame Equivalence Ratio Measurements: An Attention-Based Convolutional Neural Network Approach for Overcoming Limitations in Traditional Color Modeling"

_sensors, 2024, doi:10.3390/s24216853_

Round 1

Reviewer 1 Report

Comments and Suggestions for Authors

1、Please elaborate on how the model learns and maps the nonlinear relationship between flame image features and Φ

2、It is recommended to add a performance comparison with existing deep learning models for flame recognition or analysis tasks.

3、As the key layer for feature extraction in the ACN model, can the CP128 layer be replaced with other layers that have feature extraction capabilities?

4、What is the real-time processing capability of the ACN model? Can it handle dynamic flames or other types of flames, such as turbulent flames?

5、Can the authors explain the applicability of the model on flame images from different fuel types?

6、It is suggested to collect flame image data containing interference factors such as smoke and dust for training purposes, in order to enhance the model's ability to resist such disturbances.

Comments on the Quality of English Language

To facilitate publication and enhance readability for the audience, some potentially erroneous sentence need to be carefully reviewed.

Reviewer 2 Report

Comments and Suggestions for Authors

Authors present an Atention based convolutional neural network to calculate the flame equivalence ratio in flame images.

The paper is well writen and it is easy to follow. 

Although I am not an expert in flame burning mechanisms, in my opinion, the experiments are well conducted and the conclusions are supported by the obtained results. 

Compared to traditional methods to calculate the flame equivalence ratio, the proposed scheme obtains better results, with an accuracy of 0.996 R2.

However, I wonder why authors performs a crop and resize the original images. If you have a higher resolution camera (1280x720), why not to develope a ACN model with that size?  Although the current results are good enough, maybe with a higher resolution image, the results would be better.

Reviewer 3 Report

Comments and Suggestions for Authors

This paper addresses the in-herent limitations in traditional colour modeling, and proposes an attention-based convolutional neural network approach for overcoming limitations in traditional colour modeling, Experimental results confirm that ACNmodel surpasses existing approaches in both image analysis and combustion diag-nostics.The paper has a certain degree of innovation, but has the following shortcomings

1. The abstract of the paper should provide a more detailed description, such as numerical representation of the results.

2. Add a section 2.4 proposing the superiority of the model after "2.3 Limitations in Traditional Color Modeling".

3. The evaluation parameters in Table 4 and Table 3 should be consistent, and analyzed and explained.

4. Add two additional parameters to the change curves in Figure 13 and provide an analysis and explanation of the change curves.

5. The image in the paper did not show any preprocessing. It is suggested to add it and compare and analyze the results with and without image preprocessing.

6. The evaluation of the research results of the paper is described through four time-domain indicators, and it is recommended to add frequency-domain indicators appropriately.
